# Is a 3D-Tokenized LLM the Key to Reliable Autonomous Driving?

## Abstract

Rapid advancements in Autonomous Driving (AD) tasks turned a significant shift toward end-to-end fashion, particularly in the utilization of vision-language models (VLMs) that integrate robust logical reasoning and cognitive abilities to enable comprehensive end-to-end planning. However, these VLM-based approaches tend to integrate 2D vision tokenizers and a large language model (LLM) for ego-car planning, which lack 3D geometric priors as a cornerstone of reliable planning. Naturally, this observation raises a critical concern: **Can a 2D-tokenized LLM accurately perceive the 3D environment?** Our evaluation of current VLM-based methods across 3D object detection, vectorized map construction, and environmental caption suggests that the answer is, unfortunately, **NO**. In other words, 2D-tokenized LLM fails to provide reliable autonomous driving. In response, we introduce DETR-style 3D perceptrons as 3D tokenizers, which connect LLM with a one-layer linear projector. This simple yet elegant strategy, termed Atlas, harnesses the inherent priors of the 3D physical world, enabling it to simultaneously process high-resolution multi-view images and employ spatiotemporal modeling. Despite its simplicity, Atlas demonstrates superior performance in both 3D detection and ego planning tasks on nuScenes dataset, proving that 3D-tokenized LLM is the key to reliable autonomous driving. The code and datasets will be released.

## 1 Introduction

Autonomous Driving (AD) is a sophisticated system that integrates perception, reasoning, and planning (Janai et al., 2020; Chen et al., 2023). Perception serves as the initial stage, capturing details of the surrounding environment. This information then feeds into the reasoning component, facilitating a deeper understanding, and ultimately guiding decision-making through the planning process. Recently, the incorporation of perception, reasoning, and planning to construct end-to-end models has become prevalent. It can be broadly categorized into two distinct methodologies: modular bird's-eye view (BEV) based approaches and large vision-language model (VLM) based methods.

The modular BEV-based approaches are meticulously engineered, comprising custom-tailored modules, including 3D perception, trajectory prediction, and ego-car planning (Liang et al., 2020; Casas et al., 2021; Chen & Krähenbühl, 2022; Zhang et al., 2022; Hu et al., 2022a; Gu et al., 2023; Hu et al., 2023), as shown in Figure 1(a). While BEV representation enhances environmental perception, these methods may encounter difficulty stemming from their limited reasoning abilities. Specifically, these models tend to mimic established expert trajectories and struggle to predict multiple potential motion trajectories when confronted with novel scenarios. To tackle this challenge, VLM-based methods mark a significant turning point. They usually employ a 2D vision tokenizer (e.g., ViT-CLIP (Radford et al., 2021)) with a Large Language Model (LLM) to interpret distorted images and produce navigational commands (Xu et al., 2023; Tian et al., 2024; Wang et al., 2023b; Shao et al., 2024; Jia et al., 2023a; Sima et al., 2023). Benefiting from the robust logical reasoning and cognitive abilities of the VLM agent, the model can generate rational decisions and dialogues.

Despite the success of VLM-based algorithms, the perceptual capabilities within this paradigm is barely studied. While we argue that the perception sub-task may not be essential for end-to-end driving, the capacity to perceive the environment remains a cornerstone of reliable planning. Since VLM-based methods rely on 2D vision tokenizers for environmental perception without incorporating 3D geometric priors, an intuitive question arises: **Can a 2D-tokenized LLM accurately per-**

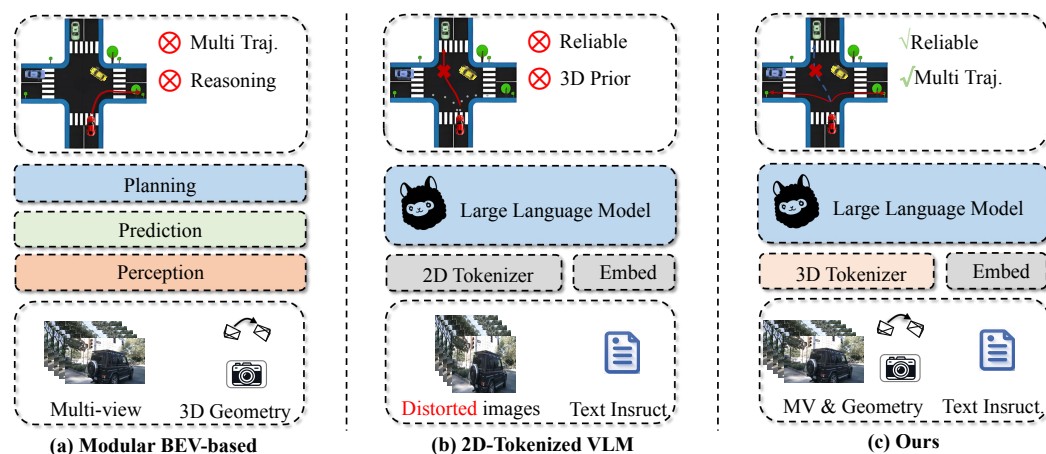

Figure 1: **Comparision among end-to-end methods**. (a) Modular BEV-based methods have three sequential modules for perception, prediction, and planning, but they cannot provide multiple potential trajectories and environment reasoning. (b) 2D-tokenized VLM projects 2D distorted images into tokens, which lack 3D prior for reliable autonomous driving. (c) Our 3D-tokenized LLM-based methods utilize 3D perceptions as 3D tokenizers, which provide potential trajectories and rich 3D priors for reliable driving.

**ceive the 3D environment?** To answer this question, we specially design experiments to evaluate the perception performance of prevalent VLM-based systems in three tasks: 3D object detection, 3D lane detection, and environmental captioning. Our findings reveal that despite extensive pre-training and expansive parameters, mainstream VLM solutions typically lag in precision when compared to specialized models designed for these tasks. This glaring gap highlights the limitations of 2D tokenizers in perceiving 3D environments.

To address this issue, we wonder if 3D vision tokenizers hold the key to Pandora. We discover that the existing DETR-style BEV framework can naturally serve as a 3D visual compression tokenizer. Therefore, we opt for the advanced StreamPETR (Wang et al., 2023a) and TopoMLP (Wu et al., 2024) as our 3D visual tokenizers, forgoing the traditional use of ViT-CLIP (Radford et al., 2021). This strategy brings three advantages: **1)** The innate priors of the 3D physical world are naturally encoded within visual tokens by introducing the position encodings. **2)** It is capable of handling high-resolution images with any aspect ratio without the risk of distorting. **3)** Video frames can be processed in a streaming manner, benefiting from DETR-style query propagation. Through evaluation of the nuScenes dataset, we demonstrate that our 3D-tokenized LLM approach achieves performance on par with specialized algorithms in tasks such as 3D object detection and lane detection.

Beyond that, we need to answer another question: **Is a 3D-tokenized LLM the key to reliable autonomous driving?** Following BEV-Planner (Li et al., 2024), we extend our exploration to the open-loop planning on the nuScenes dataset. By leveraging the 3D tokenizers for enhanced perception capabilities, our model not only comprehends the environment around the vehicle but also utilizes the LLM to formulate driving recommendations and plan the ego-car trajectory in an end-to-end manner. Remarkably, this approach eschews hand-crafted designs and achieves state-of-the-art performance on the nuScenes planning task.

In summary, our work highlights the importance of proper vision tokenizers in VLM-based AD and introduces the 3D-tokenized LLM as a solution. We showcase its superiority in adeptly addressing challenges across multiple tasks such as 3D perception, vectorized map construction, environmental caption, and planning within autonomous driving systems. Our model demonstrates superior performance in both benchmark evaluations and practical downstream applications, proving its reliability and versatility. Furthermore, our framework paves the way for pioneering end-to-end LLM-driven solutions in autonomous driving, potentially transforming how these systems are developed.

## 2 CAN A 2D-TOKENIZED LLM PERCEIVE 3D ENVIRONMENT?

Current VLM-based methods (Xu et al., 2023; Tian et al., 2024; Wang et al., 2023b; Shao et al., 2024; Jia et al., 2023a) in AD tend to employ 2D vision tokenizers. They operate without incorporating

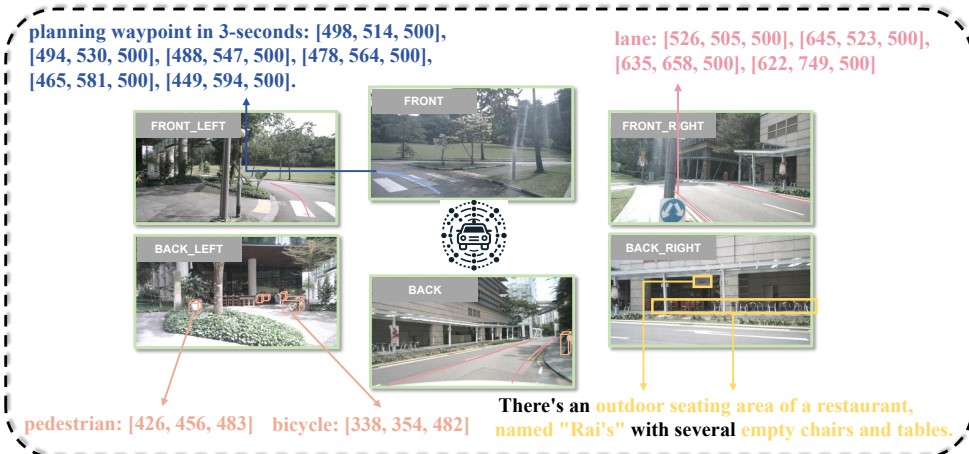

Figure 2: **Brief answer format of datasets.** It transforms several tasks, such as 3D object detection, map perception, environment caption, and ego-car planning, into a uniform text format. We discretize the bird's-eye view (BEV) space, spanning from -50 meters to +50 meters, into 1,000 bins.

geometric 3D priors, raising concerns about their capability to accurately perceive and describe 3D environments, which is crucial for reliable planning. In this section, we provide insightful analysis and reveal the limitations of relying solely on 2D tokenizers for understanding 3D driving scenes, including 3D perception and visual captioning.

## 2.1 2D-TOKENIZED LLM FOR PERCEPTION

To investigate the 3D understanding capability of current VLM-based approaches, we first conduct experiments on traditional perception tasks: 3D object detection and 3D lane detection. In this part, we introduce datasets, models, and metrics.

**Datasets.** We design datasets tailored for VLM methods built upon popular multi-view benchmark nuScenes (Caesar et al., 2020), as shown in Figure 2. For the 3D detection task, we construct question-and-answer (QA) pairs that focus on pinpointing the locations of objects surrounding the ego vehicle. Each question prompts the model to extract spatial information about the target objects from six views. The corresponding answers require the model to identify both the category and the 3D coordinates of objects. Similarly, the dataset for 3D lane detection also comprises QA pairs, whose answers are lane points borrowed from OpenLane-V2 subset-B (Wang et al., 2024). Here, each road is depicted using four consecutive points describing the road centerline. More details can be found in the supplementary.

**Models.** All 2D-tokenized LLMs in our study adhere to a uniform architecture, which consists of three main components: 2D tokenizer, projector, and large language model. The 2D tokenizer follows ViT-CLIP (Radford et al., 2021) to extract visual features from multiple perspectives of images. For the projection module, we incorporate a single convolutional layer to bridge the 2D tokenizer and LLM. Besides, we utilize diverse pre-trained LLMs, such as LLaMA (Touvron et al., 2023), LLaVA (Liu et al., 2024), Vicuna (Chiang et al., 2023), which are comprehensive processing of complex visual information to generate the perception of the environment, to prove consistency and fairness in our exploration. Additionally, another available VLM-based model pre-trained on 2D object detection Merlin (Yu et al., 2023) is also evaluated.

**Metrics.** In this study, we employ the F1 score as the main evaluation metric. The choice of the F1 score is motivated by two primary considerations: First, VLMs cannot deliver the necessary predictive confidence for metrics such as mean Average Precision (mAP). Second, traditional perceptual metrics commonly encourage numerous redundant predictions, which can clutter the model output. In contrast, VLMs are designed to generate more targeted and focused predictions, making the F1 score a better fit for assessing these models. In this work, for 3D detection, we choose threshold distances of 0.5, 1.0, 2.0, and 4.0 meters to define positive predictions, similar to the discrimina-

Table 1: Comparisons with task-specific and VLM-based methods for 3D object detection tasks using our proposed dataset. The bold **numbers** represent the highest accuracy achieved in each category. The $P_k$, $R_k$, and $F1_k$ represent the Precision, Recall, and respective F1 score ultimate $k$ as threshold distances to define positive prediction. The Spe. represents task-specialist model.

| | Method | Tokenizers | $P_{0.5}$ | $R_{0.5}$ | $F1_{0.5}$ | $P_{1.0}$ | $R_{1.0}$ | $F1_{1.0}$ | $P_{2.0}$ | $R_{2.0}$ | $F1_{2.0}$ | $P_{4.0}$ | $R_{4.0}$ | $F1_{4.0}$ |
|---|---|---|---|---|---|---|---|---|---|---|---|---|---|---|
| Spe. | PETR (Liu et al., 2022) | - | 12.4 | 21.5 | 15.8 | 20.0 | 30.5 | 24.1 | 27.5 | 37.7 | 31.8 | 33.8 | 42.6 | 37.7 |
| | StreamPETR (Wang et al., 2023a) | - | **22.7** | **41.3** | **29.3** | **31.6** | **49.5** | **38.6** | **38.1** | **54.2** | **44.7** | **42.5** | **56.9** | **48.7** |
| VLM | LLaMA (Touvron et al., 2023) | 2D | 0.3 | 1.1 | 0.4 | 0.6 | 2.6 | 1.0 | 1.5 | 5.8 | 2.4 | 3.5 | 12.8 | 5.5 |
| | LLaVA (Liu et al., 2024) | 2D | 2.0 | 20.3 | 3.0 | 3.6 | 35.7 | 6.5 | 6.5 | 50.3 | 11.6 | 10.9 | 62.8 | 18.9 |
| | Vicuna (Chiang et al., 2023) | 2D | 2.0 | 20.1 | 2.5 | 2.9 | 35.6 | 5.4 | 5.9 | 51.1 | 10.1 | 9.4 | 63.8 | 16.4 |
| | Merlin (Yu et al., 2023) | 2D | 3.0 | 22.5 | 5.3 | 4.1 | 36.1 | 7.4 | 6.6 | 52.6 | 11.7 | 12.1 | 64.3 | 20.4 |
| | **Atlas**(Ours) | **3D** | 15.0 | 61.2 | 24.1 | 27.2 | 74.0 | 39.8 | 36.2 | 79.2 | 49.7 | 41.2 | 81.2 | 54.6 |

tion levels used in detection mAP calculations. As for 3D lane detection, we follow OpenLane-V2 evaluation protocol (Wang et al., 2024) to compute the F1 score.

**3D Object Detection.** In this study, we conduct extensive experiments to evaluate the performance of VLMs on 3D detection, as listed in Table 1. As a comparison, Table 1 also includes task-specific models such as PETR (Liu et al., 2022) and StreamPETR (Wang et al., 2023a). Among these, the state-of-the-art detector StreamPETR achieves an $F1_{4.0}$ score of 48.7. Despite the rich contextual knowledge and extensive parameters, 2D-tokenized LLM methods exhibit a considerable performance drop in both precision and recall, leading to surprisingly low F1 scores. These methods struggle with detecting objects in the vicinity of the ego vehicle, highlighting a considerable disparity in 3D object detection capabilities between VLM-based methods and dedicated task-specific approaches.

**3D Lane Detection.** Vectorized maps provide a driving route for ego car, serving as a crucial perception task for autonomous driving. We present experiments of a state-of-the-art task-specific model TopoMLP (Wu et al., 2024) and several aforementioned 2D-tokenized LLM methods on lane detection. The main results are shown in Table 2.

Table 2: 3D lane detection.

| Method | Tokenizers | P | R | F1 |
|---|---|---|---|---|
| TopoMLP (Wu et al., 2024) | - | 50.6 | 55.7 | 53.0 |
| LLaVA (Liu et al., 2024) | 2D | 10.4 | 9.8 | 10.0 |
| Vicuna (Chiang et al., 2023) | 2D | 11.7 | 10.3 | 10.9 |
| Merlin (Yu et al., 2023) | 2D | 22.1 | 22.4 | 22.2 |
| **Atlas**(ours) | **3D** | 45.7 | 39.1 | 42.2 |

Similarly, the performance of 2D-tokenized LLM methods is far away from the task-specific model, struggling to deal with 3D lane detection.

## 2.2 2D-TOKENIZED LLM FOR CAPTIONING

In addition to basic environmental perception tasks, LLMs can be adapted to perform more complex tasks like extracting and interpreting key features from visual for captioning environments. This capability extends the utility of LLMs in practical applications, and leverages world knowledge and reasoning ability, particularly in scenarios requiring detailed environmental understanding.

To explore whether a 2D-tokenized LLM could serve as an effective perceptron, we develop a specialized version of the model for environmental captioning. This variant utilizes Vicuna (Chiang et al., 2023) as its underlying LLM, tasked with capturing and describing the environment of ego vehicle. This description includes various elements such as the location and quantity of nearby vehicles and pedestrians, traffic dynamics, concerning surrounding lanes of pedestrian crossing and road.

Despite the advanced capabilities of VLMs in generating natural language descriptions, as illustrated in Figure 3, our findings indicate that the 2D-tokenized LLM struggles with accurate environmental perception. The model frequently produces erroneous or "hallucinated" descriptions, which suggests that it still falls short of reliable perception in practical applications. This underscores the challenges and limitations inherent in deploying LLMs for complex perceptual tasks in dynamic environments.

**Remark.** To sum up, the experiments above reveal a significant limitation in the perception capabilities of LLMs that rely on 2D visual tokenizers. This limitation poses serious challenges for reliable ego vehicle planning. We claim that the primary reason for this limitation lies in the inability of 2D visual tokenizers to effectively integrate 3D spatial priors. To address the limitation, we introduce advanced pre-trained 3D perception models as 3D tokenizers in the following section.

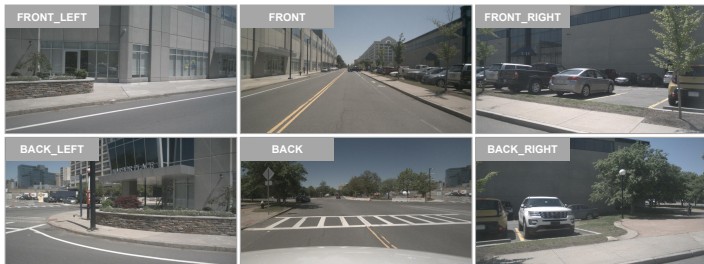

**Question:** Communicate a narrative of the setting within **FRONT** view image.
**GT explanation**: The parked vehicles vary, including sedans and larger vehicles like SUVs and pickup trucks. There is one pedestrian walking on the sidewalk to the left of the image, but one moving vehicle is visible on the street itself, which suggests that the traffic is not heavy at this moment.

**Responser:** Vicuna
**Answer:** The parked vehicles are uniform, consisting of compact cars and **motorcycles**. There are **two** pedestrians on the sidewalk to the left of the image, and **several moving vehicles** are racing down the street, which suggests that the traffic is extremely **heavy** at this moment.

**Responser:** Atlas
**Answer:** The parked vehicles include a variety, such as **sedans and larger types like SUVs and pickups.** A **single** pedestrian is observed walking along the sidewalk on the left of the photo, and there is **one vehicle in motion** on the road itself, pointing to **light** traffic conditions currently.

**Question:** Convey the story depicted in the **BACK_LEFT** view photograph.
**GT explanation**: There are a few vehicles parked in the background, but there is no immediate traffic flow or movement captured in this frame. The vehicles present in the background include a few parked cars and a FedEx truck. The road markings include a pedestrian crossing, indicated by the parallel lines on the road.

**Responser:** Vicuna
**Answer:** There are **several vehicles** parked in the background, but a slight traffic flow is captured in this frame. In the background, the vehicles include **a large FedEx truck.** The road markings include a pedestrian crossing.

**Responser:** Atlas
**Answer:** **A few vehicles** are parked in the background, with no significant traffic flow or movement evident in this frame. In the background, the scene includes **a couple of parked cars and a FedEx truck.** Road markings present include a pedestrian crossing.

Figure 3: **Comparsion between 2D-tokenized and our 3D-tokenized VLMs on driving caption.** The 2D-tokenized VLM sometimes generates "hallucinated" descriptions, while our 3D-tokenized VLM is able to produce accurate and comprehensive captions for driving environment.

## 3 3D-TOKENIZED LLM FOR RELIABLE AUTONOMOUS DRIVING

### 3.1 3D-TOKENIZED LLM

Distinct from 2D-tokenized LLM, we introduce 3D tokenizers founded upon a DETR-inspired architecture into LLM, formulating a 3D-tokenized LLM framework, named Atlas. In specific, Altas consists of three primary components. Initially, the model employs 3D tokenizers, StreamPETR (Wang et al., 2023a) and TopoMLP (Wu et al., 2024), to process multi-view images into DETR-style query representations. Following this, these queries are streamlined through a single linear layer, functioning as a projector, to align with the LLM. The final component of Atlas is an LLM, designed as Vicuna (Chiang et al., 2023). This approach brings significant benefits in *incorporating 3D innate prior, achieving high resolution, and facilitating temporal propagation*, as previously elaborated.

**3D Environment Perception.** The performance of Atlas is evaluated on standard datasets tailored to the tasks of 3D object detection and 3D lane detection, as reported in Table 1 and Table 2. The results demonstrate that 3D-tokenized LLM achieves remarkable performance across both tasks. Besides, 3D-tokenized LLM performs better than 2D-tokenized LLM on driving environment captioning, as shown in Figure 3, thereby affirming the significant advantages of utilizing 3D tokenizers. In addition to representing 3D environment, our ultimate goal is to achieve reliable autonomous driving. In the following, we will evaluate the performance of 3D-tokenized LLM on ego-car planning.

### 3.2 IMPLEMENTATION

The whole model trains with 8 Tesla A100 GPUs, with training times of approximately 100 hours.

**Dataset.** We employ the nuScenes planning dataset (Caesar et al., 2020) in our experiments of reliable autonomous driving. As illustrated in Figure 2, we have reformatted the planning data into a question-answer format to facilitate our analysis. Previous research (Li et al., 2023b) has established that the "ego states"—sensor-provided data on the autonomous vehicle such as velocity, acceleration, yaw angle, and historical trajectory—play a crucial role in open-loop planning. Additionally,

Table 3: **Comparisons on the planning.** For a fair comparison, we refer to the reproduced results in BEV-Planner (Li et al., 2023b). The bold **numbers** represent the highest accuracy.

| Method | High-level Command | Ego States Bev | Planner | L2 (m) 1s | 2s | 3s | Avg. | Collision (%) 1s | 2s | 3s | Avg. |
|---|---|---|---|---|---|---|---|---|---|---|---|
| FF (Hu et al., 2021) | ✗ | ✔ | ✔ | 0.55 | 1.20 | 2.54 | 1.43 | 0.06 | 0.17 | 1.07 | 0.43 |
| ST-P3 (Hu et al., 2022b) | ✔ | ✗ | ✗ | 1.59 | 2.64 | 3.73 | 2.65 | 0.69 | 3.62 | 8.39 | 4.23 |
| | ✔ | ✔ | ✔ | 1.33 | 2.11 | 2.90 | 2.11 | 0.23 | 0.62 | 1.27 | 0.71 |
| UniAD (Hu et al., 2023) | ✔ | ✗ | ✗ | 0.59 | 1.01 | 1.48 | 1.03 | 0.16 | 0.51 | 1.64 | 0.77 |
| | ✔ | ✔ | ✔ | 0.20 | 0.42 | 0.75 | 0.46 | 0.02 | 0.25 | 0.84 | 0.37 |
| VAD-Base (Jiang et al., 2023) | ✔ | ✗ | ✗ | 0.69 | 1.22 | 1.83 | 1.25 | 0.06 | 0.68 | 2.52 | 1.09 |
| | ✔ | ✔ | ✔ | 0.17 | 0.34 | 0.60 | 0.37 | 0.04 | 0.27 | 0.67 | 0.33 |
| Ego-MLP (Zhai et al., 2023a) | ✔ | ✗ | ✔ | **0.15** | 0.32 | 0.59 | 0.35 | **0.00** | 0.27 | 0.85 | 0.37 |
| BEV-Planner (Li et al., 2023b) | ✔ | ✗ | ✗ | 0.30 | 0.52 | 0.83 | 0.55 | 0.10 | 0.37 | 1.30 | 0.59 |
| | ✔ | ✔ | ✔ | 0.16 | 0.32 | 0.57 | 0.35 | **0.00** | 0.29 | 0.73 | 0.34 |
| LLaVA (Liu et al., 2024) | ✔ | ✗ | ✗ | 1.04 | 1.74 | 2.57 | 1.79 | 0.58 | 1.17 | 1.74 | 1.16 |
| Vicuna (Chiang et al., 2023) | ✔ | ✗ | ✗ | 1.06 | 1.80 | 2.54 | 1.80 | 0.60 | 1.21 | 1.78 | 1.20 |
| Merlin (Yu et al., 2023) | ✔ | ✗ | ✗ | 1.03 | 1.71 | 2.40 | 1.71 | 0.48 | 1.05 | 1.77 | 1.10 |
| Atlas | ✗ | ✗ | ✗ | 1.69 | 1.89 | 2.25 | 1.94 | 0.51 | 0.85 | 1.44 | 0.93 |
| | ✔ | ✗ | ✗ | 0.52 | 0.97 | 1.53 | 1.00 | 0.15 | 0.31 | 0.70 | 0.38 |
| | ✔ | ✔ | ✔ | 0.18 | **0.21** | **0.26** | **0.21** | 0.12 | **0.13** | **0.16** | **0.13** |

to aid in navigation, especially at intersections, it is essential to incorporate a high-level command (e.g., go straight, turn left, turn right) which provides directional guidance. Building on these insights, we propose the question-and-answer pairs demand the models to predict future velocity and acceleration based on the current state and to subsequently generate planning waypoints for the ego-car prompting by a high-level command. This processing called chain-of-thought (Wei et al., 2022), not only enhances the interpretability of the model's reasoning process but also its reliability. A typical example is shown in Figure 2, and additional details about the dataset are available in the supplementary materials.

**Metrics.** We adhere to standard practices by utilizing the implementation provided by ST-P3 (Hu et al., 2022b) to assess planning over time horizons of 1s, 2s, and 3s. We assess the performance with two widely accepted metrics: the L2 error calculated by comparing the predicted trajectories of the ego vehicle with the ground-truth trajectories at corresponding waypoints, and the collision rate calculated by checking for any intersections between the ego vehicle and other entities.

## 3.3 MAIN RESULTS

In this section, we evaluate the performance of Atlas, by comparing it against existing state-of-the-art BEV-based planners, as detailed in Table 3. Our experimental results reveal that Atlas achieves substantial improvements over the SoTA methods, reducing the average L2 metric by 40.0% and the average Collision metric by 60.6%. These significant enhancements corroborate the effectiveness of the 3D-tokenized LLMs, which we consider as the key to reliable autonomous driving.

Further, to ascertain whether the performance improvements are solely attributable to the inclusion of ego state information—a frequent topic of discussion within the community—we conduct additional experiments by removing the ego state data during both training and testing. In this experimental setting, compared to the prevailing VLM-based methods, our Atlas demonstrates superior performance and robustly validates the effectiveness of 3D tokenizers. Despite this, Atlas continues to outperform other BEV-based methods in terms of collision rates. However, the performance on the L2 metric is comparable to other methods. We hypothesize that this outcome may stem from the inherent capabilities of the LLM to predict multiple potential motion trajectories and make rational decisions, which, while confronted with novel scenarios, deviate from the ground truth.

## 3.4 ABLATION STUDY

To avoid unnecessary misunderstandings, our ablation does not introduce any ego states.

Table 4: A set of ablative studies on 3D object detection and ego-car planning. The adopted algorithm designs and hyper-parameter settings are marked in **bold** . See §3.4 for details.

|  | 3D Detection | | Planning | |
|---|---|---|---|---|
|  | $F1_{1.0}$ | $F1_{2.0}$ | Avg. L2 | Avg. Col. |
| Vicuna | 5.4 | 10.1 | 2.19 | 2.75 |
| +QR | 30.7 | 41.2 | 1.22 | 0.62 |
| + RP | 34.6 | 46.5 | 1.10 | 0.44 |
| **+ MQ** | **39.8** | **49.7** | **1.00** | **0.38** |

(a) Component Effect. QR, RP, and MQ mean Query Representation, Reference Point embedding, and Memory Queue.

| PT | SP | TM | Avg. L2 | Avg. Col. |
|---|---|---|---|---|
| ✔ | - | - | 1.51 | 1.05 |
| - | ✔ | - | 1.06 | 0.41 |
| **-** | **✔** | **✔** | **1.00** | **0.38** |

(b) Effect of different 3D tokenizers. PT, SP and TM represent PETR, StreamPETR and TopoMLP.

| Resolution | Avg. L2 | Avg. Col. |
|---|---|---|
| 336×336 | 1.66 | 0.94 |
| 320×800 | 1.41 | 0.58 |
| **800×1600** | **1.00** | **0.38** |

(c) Effect of resolution.

| RP. emb. | Avg. L2 | Avg. Col. |
|---|---|---|
| none | 1.18 | 0.57 |
| sin-cos | 1.21 | 0.57 |
| learned | 1.19 | 0.56 |
| **RP** | **1.00** | **0.38** |

(d) Reference point embeddings.

| LLMs | Avg. L2 | Avg. Col. |
|---|---|---|
| LLaMA (Touvron et al., 2023) | 1.14 | 0.47 |
| LLaVA (Liu et al., 2024) | 1.03 | 0.39 |
| **Vicuna** (Chiang et al., 2023) | **1.00** | **0.38** |
| Merlin (Yu et al., 2023) | **0.99** | 0.42 |

(e) Different pretrained LLMs.

**Component Effect.** We conduct ablation studies to analyze our proposed 3D-tokenized LLM, considering several key aspects: *query representation*, *reference point embedding*, and *memory queue*, all decoupling from StreamPETR on 3D detection and planning. The results are summarized in Table 4a. Our experiments demonstrate that each component progressively enhances performance in both tasks. Furthermore, we observe a synergistic effect where improvements in one task appear to amplify accuracy in the other, strongly proving that the capacity to perceive the environment remains a cornerstone of reliable planning.

**3D Tokenizers.** We investigate the effectiveness of various 3D tokenizers for ego-car planning, which are the central enhancements introduced in our study. The results are shown in Table 4b. The tokenizers we evaluate include PETR (PT) (Liu et al., 2022), StreamPETR (SP) (Wang et al., 2023a), and TopoMLP (TM) (Wu et al., 2024). Our incorporation of progressively advanced 3D perceptrons into LLM demonstrates a notable improvement in planning performance, underscoring the significance of 3D perception in achieving robust autonomous driving. Furthermore, we integrate TopoMLP to provide supplementary lane line information. This addition results in a modest enhancement in performance, suggesting the potential benefits of incorporating contextual roadway features into the motion planning process.

**Resolution.** Our approach integrates 3D tokenizers with adjustable image resolution capabilities, which aligns well with real-world applications in autonomous driving. As Table 4c presents, we observe that increasing the image resolution leads to a noticeable improvement in performance. This evidence indicates that our method holds significant advantages over traditional VLM techniques, particularly in terms of flexibility and efficacy in handling diverse image resolutions.

**Reference Point Embeddings.** Our Atlas introduces an important concept: 3D tokenizers equipped with reference point embeddings, following the setting of StreamPETR (Wang et al., 2023a) and TopoMLP (Wu et al., 2024). Here, we evaluate the model performance of decoupling reference point embedding and query embedding. Our initial approaches relied solely on query representations (i.e., "none" in Table 4d), which overlooks the crucial 3D spatial context—termed as the reference point. However, as shown in Table 4d, simply applying conventional embedding techniques (Carion et al., 2020), like sin-cos position embedding and learned position embedding, to 3D queries do not markedly influence performance. This outcome underscores the unique advantages of reference points. To effectively utilize this, we incorporate offset mappings from the reference points via a single layer projector aka reference point embeddings to the 3D query representation (i.e., "RP" in Table 4d). Notably, this method achieves remarkable improvements in accuracy.

**Pretrained LLMs.** In our experiments, we evaluate different LLMs that varied in their pre-training methodologies, as detailed in Table 4e. Our results show that LLMs pre-trained with methods that align text and images significantly outperform others in planning tasks. We attribute this enhanced performance to the multimodal nature of their training. Additionally, our analysis reveals that mod-

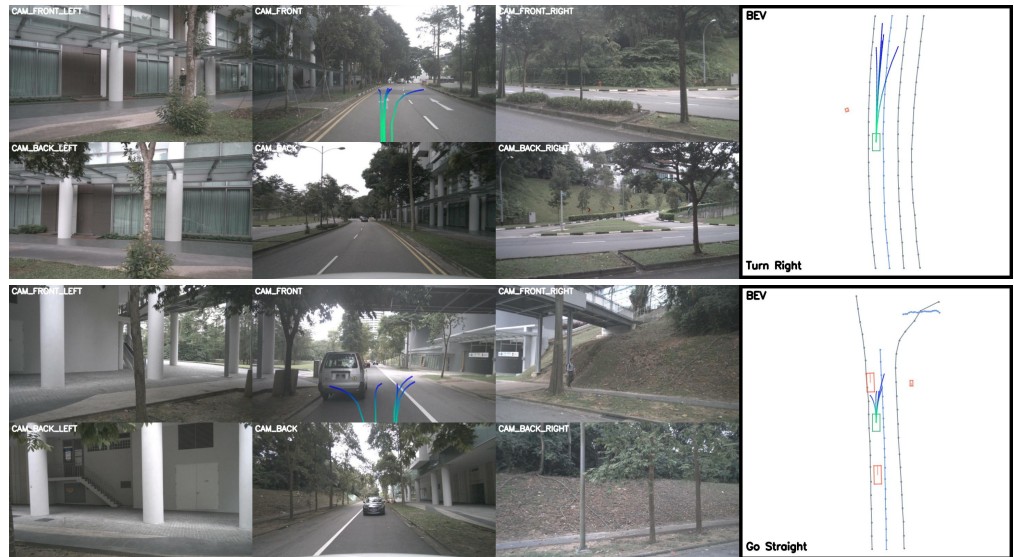

Figure 4: **Qualitative results with diverse planning from Atlas.** The five planning trajectories presented here are generated through five iterations of utilizing our 3D-tokenized LLM. It is obvious that Atlas is able to output different potential planning trajectories thanks to LLM.

els pre-trained with various visual-language data exhibited no significant differences in planning performance. We believe this is due to the absence of 3D data in their pre-training processes, suggesting that the inclusion of 3D data in pre-training, as 3D tokenizers do, is necessary.

**Chain of Thought (CoT).** In the realm of autonomous driving, recent works (Zhai et al., 2023a; Li et al., 2023b) converge on a key insight: the state of the ego vehicle is a pivotal factor in shaping open-loop planning strategies. To this end, we delineate the ego states into four distinct yet interrelated dimensions: velocity (V), acceleration (A), yaw angle (Y), and the historical trajectory (T). To evaluate the influence of each dimension, we conduct ego planning based on the predicated of these parameters. The experimental findings, as outlined in Table 5, where "P" denotes "Planning". No-

Table 5: Effective of the CoT.

| chain | Avg. L2 | Avg. Col. |
|---|---|---|
| P | 1.33 | 0.79 |
| V-P | 1.21 | 0.61 |
| **V-A-P** | **1.00** | **0.38** |
| V-A-Y-P | 1.15 | 0.55 |
| V-A-T-P | 1.40 | 0.81 |
| P-V-A | 1.01 | 0.40 |

tably, our results diverge from prevailing research, indicating that the yaw angle and historical trajectory do not enhance the efficacy of the planning process. This counterintuitive outcome is likely a consequence of the inherent difficulties in the precise forecasting of these variables (Wei et al., 2023; Bai et al., 2024). Moreover, we discover an interesting aspect of our model's robustness: the sequence in which these parameters are predicted does not impact the performance. This suggests that altering the order of prediction (e.g., reversed) does not increase computational time.

### 3.5 QUALITATIVE RESULTS

We also conduct a qualitative analysis by visualizing the trajectory predictions made by Atlas, as shown in Figure 4. We execute the 3D-tokenized LLM five times to produce five depicted planning trajectories. The results demonstrate that Atlas is capable of generating multiple feasible plans for autonomous driving that are not only practical but also adhere to safety standards. Specifically, Atlas successfully devises various potential routes tailored to distinct driving scenarios, including following other vehicles, lane changing, and overtaking. Importantly, Atlas effectively identifies and avoids pedestrians and cars, showcasing its robust capability in ensuring road safety.

### 4 RELATED WORK

**DETR-style BEV Perception.** DETR (Carion et al., 2020) is initially proposed to address the challenge of end-to-end detection, and further extensively applied in BEV perception (Liu et al., 2022; Li et al., 2022; Liu et al., 2023a; Lin et al., 2022), thereby significantly advancing its development.

DETR3D (Wang et al., 2022) is a pioneering work that introduces the concept of *3D object queries*, which interact with multi-view image features to produce sparse yet informative object representations. Further, PETR (Liu et al., 2022; 2023b) introduces the concept of 3D position encoding, and BEVFormer (Li et al., 2022) brings BEV temporal modeling. StreamPETR (Wang et al., 2023a) and Sparsev2 (Lin et al., 2023) use object queries as a vessel for temporal modeling, effectively propagating temporal information while achieving SoTA performance with commendable efficiency. *In a notable finding within StreamPETR, the inclusion of additional multi-frame image feature interactions does not enhance performance, suggesting that the highly compressed object queries are sufficiently expressive to encapsulate all necessary information for BEV perception.* Moreover, the application of DETR framework has been expanded to *map queries* by works such as MapTR (Liao et al., 2023), TopoNet (Li et al., 2023a) and TopoMLP (Wu et al., 2024), which are instrumental in the construction of vectorized map representations.

**BEV-based End-to-end Driving.** Traditional autonomous driving systems have often relied on manual rules, which can be cumbersome and complex, struggling to cover the numerous corner cases. In recent years, there has been a pronounced shift towards end-to-end autonomous driving approaches, which have demonstrated significant progress in simplifying and streamlining the pipeline. UniAD (Hu et al., 2023) is a pioneering work that introduces an end-to-end framework encompassing tasks such as perception, prediction, and planning, with these tasks executed sequentially to ultimately produce control outputs. Building upon this framework, VAD (Alexanian et al., 1990) further streamlines the pipeline, enhancing efficiency and reducing complexity. However, AD-MLP (Zhai et al., 2023b) and BEV-Planner (Li et al., 2024) have observed that existing end-to-end methods can achieve high performance on open-loop benchmarks like nuScenes (Caesar et al., 2020) by simply fitting to the ego status of the autonomous vehicle. This finding suggests that the integration of planning and control in these models may not yet fully capture the complexities of real-world driving scenarios. Subsequent works, such as Think-Twice (Jia et al., 2023b) and VADv2 (Chen et al., 2024), have made substantial advancements in closed-loop simulators like Carla (Dosovitskiy et al., 2017). Following BEV-Planner (Li et al., 2024), we present results with and without the ego status to address the open-loop challenges on the nuScenes (Caesar et al., 2020).

**VLM-Agent for Autonomous Driving.** The visual-language model (VLM) demonstrates promising results in the fields of visual-language understanding and logical reasoning, and has been extended to autonomous driving (Xu et al., 2023; Tian et al., 2024; Wang et al., 2023b; Shao et al., 2024; Jia et al., 2023a; Xie et al., 2024). DriveGPT4 (Xu et al., 2023) employs a VLM model to predict driving commands and provide rational explanations for its decisions. DriveLM (Sima et al., 2023) excels at conversing about environmental information, while ADriver-I (Jia et al., 2023a) focuses on predicting low-level vehicle signals. Furthermore, DriveMLM (Wang et al., 2023b) and LMDrive (Shao et al., 2024) have implemented end-to-end autonomous driving solutions and validated effectiveness on CARLA (Dosovitskiy et al., 2017) closed-loop benchmarks, showcasing the potential of VLM-based agents. Despite impressive progress, no work explores how the 3D-tokenized LLM influences real-life autonomous driving.

## 5 CONCLUSION

In this paper, we explored VLM-based methods increasingly used in autonomous driving, focusing first on perception. We found large gaps between task-specific and 2D tokenized LLM-based methods in environmental perception, which is essential for reliable autonomous driving. To address these gaps, we introduced Atlas, a system combining DETR-style 3D perceptrons with LLMs. This approach integrates 3D priors for better depth perception and supports high-resolution, multi-view images, and temporal modeling through query propagation. Our evaluation of Atlas on nuScenes dataset revealed substantial improvements in 3D detection and planning, surpassing established methods. This confirms our belief that 3D-tokenized LLM is the key to reliable autonomous driving.

**Limitations.** This paper aims to demonstrate the effectiveness of the 3D tokenizer for VLM-based autonomous driving. Although our method has demonstrated outstanding performance in open-loop planning, it has not yet been tested on a closed-loop dataset. However, existing close-loop benchmarks (e.g., CARLA (Dosovitskiy et al., 2017)) lack reality, which fails to verify our motivation. Moreover, this paper lacks of performance comparison with VLM-based AD methods. This omission is due to the proprietary codes for these methods.

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
