# Appendix

## A  DATASETS DETAILS

To align with the requirements of VLM-based models, the necessary step is to transform all evaluation datasets into a textual format, specifically structured as question-answer pairs. In this section, we will delve deeper into the specifics of various datasets, including 3D object detection (§A.1), 3D lane detection (§A.2), driving captioning (§A.3), and ego planning (§A.4).

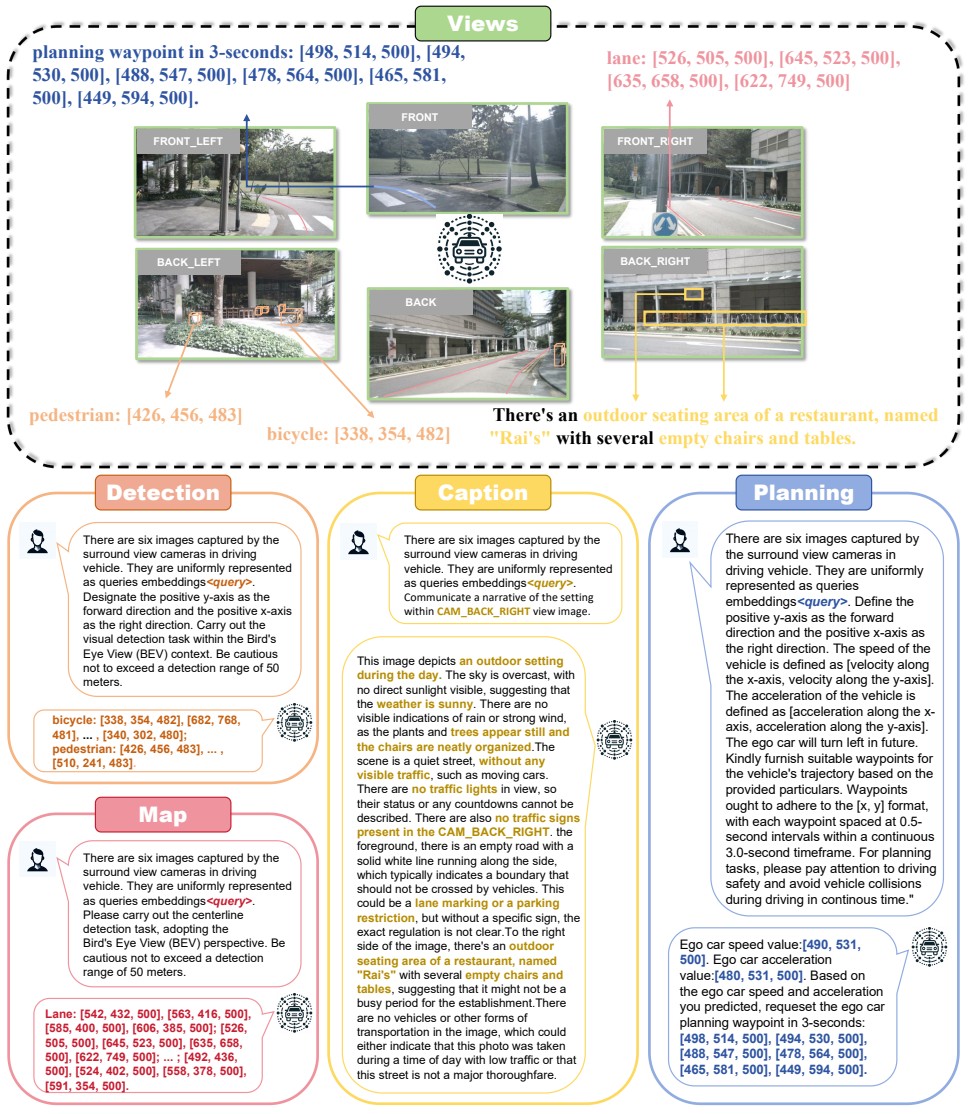

Figure 1: **Our constructed question-answer pairs for VLM-based methods.** It transforms several critical driving reasoning tasks, such as 3D object detection, map perception, environment caption, and ego-car planning, into a uniform text format.

## A.1 3D OBJECT DETECTION

The 3D object detection utilized in the VLM-based method (2D/3D visual tokenizers with LLM) evaluation is based on nuScenes (Caesar et al., 2020). To adapt to the inputs and outputs of LLM, we convert the detection task into a text-format question-answer task. Here, the question is randomly sampled from a pool that is listed in Table 1. As seen, we set a special token '<query>' to accept tokens from 3D tokenizers. If the inputs are six-view images, we replace the text 'They are uniformly represented as queries embeddings<query>' in question with 'They represent left rear image<query>, left front image<query>, direct front image<query>, right front image<query>, right rear image<query>, and direct rear image<query>.'. As for the answer, we choose the category name and 3D center points of each bounding box, as shown in Figure 1. To facilitate more efficient localization, we discretize the bird's-eye view (BEV) space ranging from -50 meters to +50 meters into 1,000 bins.

---

- "There are six images captured by the surround view cameras in driving vehicle. They are uniformly represented as queries embeddings<query>. Define the positive y-axis as the forward direction and the positive x-axis as the right direction. Please complete the visual detection task under the Bird's Eye View (BEV) perspective. Ensure that the detection range does not exceed 50 meters."

- "There are six images captured by the surround view cameras in driving vehicle. They are uniformly represented as queries embeddings<query>. Establish the positive y-axis as the frontward direction and the positive x-axis as the rightward direction. Kindly execute the visual detection task within the Bird's Eye View (BEV) framework. Be mindful not to exceed a detection range of 50 meters."

- "There are six images captured by the surround view cameras in driving vehicle. They are uniformly represented as queries embeddings<query>. Set the forward direction as the positive y-axis and the right direction as the positive x-axis. Please carry out the visual detection task within the Bird's Eye View (BEV) context. Ensure that the detection range remains within 50 meters."

- ......

---

Table 1: **Question pool of 3D object detection for VLM-based methods.**

## A.2 3D LANE DETECTION

We formulate a 3D lane detection dataset with question-answer pairs based on the OpenLane-V2 Subset-B (Wang et al., 2024), which itself originates from the nuScenes dataset. A representative is shown in Figure 1. Their questions are sampled from Table 2, and the corresponding answer comprises a set of four lane points. Analogous to the 3D object detection dataset, we discretize the BEV space, spanning from -50 meters to +50 meters, into 1,000 bins.

---

- "There are six images captured by the surround view cameras in driving vehicle. They are uniformly represented as queries embeddings<query>. Please complete the centerline detection task under the Bird's Eye View (BEV) perspective. Ensure that the detection range does not exceed 50 meters."

- "There are six images captured by the surround view cameras in driving vehicle. They are uniformly represented as queries embeddings<query>. Be mindful not to exceed a detection range of 50 meters."

- "There are six images captured by the surround view cameras in driving vehicle. They are uniformly represented as queries embeddings<query>. Could you complete the task of detecting the centerline from the Bird's Eye View (BEV) perspective? Ensure that the detection range remains within 50 meters."

- ......

---

Table 2: **Question pool of 3D lane detection for VLM-based methods.**

## A.3 Driving Captioning

Our driving captioning dataset is created through the annotation of nuScenes, leveraging the capabilities of GPT-4V. The specific prompt utilized in GPT-4V is detailed in Table 3, while an illustrative example is presented in Figure 1. It is worth mentioning that, to harness the full potential of GPT-4V, we request a unique description for each individual view, resulting in a total of approximately 180k question-answer pairs.

> • "Describe the current traffic conditions. If there are traffic lights in the image, describe the status of all the traffic lights, including any countdowns; if there are none, please do not respond. If there are traffic signs in the picture, identify and explain each one; if there are none, no explanation is necessary. If there are other vehicles in the picture, describe them in more detail. Please ensure the answer does not exceed 600 words. Answers must be in English."

Table 3: **Prompt used in GPT-4V for caption generation.**

## A.4 Ego Planning

Similar to 3D object and lane detection, we adapt the nuScenes dataset into a question-answer pairs format. Following the chain-of-thought approach, we prompt our model to generate safe driving plans and to describe various ego states, such as velocity and acceleration. The specific questions used are sampled from Table 4. For the answers, the model predicts the current state's velocity and acceleration and then generates the ego-car's planning waypoints for the next 3 seconds at 0.5-second intervals. This approach mirrors our methods in 3D object detection and 3D lane detection, where we discretize the BEV space, which ranges from -50 to +50 meters, into 1,000 bins. Similarly, we discretize both velocity and acceleration across a range from -50 m/s (m/s$^2$) to +50 m/s (m/s$^2$) into 1,000 bins each.

# B  Model Details

## B.1 3D Tokenizers Pre-training

We pre-train two distinct 3D tokenizers: StreamPETR (Wang et al., 2023) and TopoMLP (Wu et al., 2024). StreamPETR (Wang et al., 2023) is designed for multi-view 3D object detection. We utilize a ViT-L backbone (Fang et al., 2023) and process images at a high resolution of 800x1600. Moreover, we follow the official training schedule established for the nuScenes dataset. TopoMLP (Wu et al., 2024) focuses on constructing vectorized maps from multiple views. To maintain methodological consistency with StreamPETR, we employ the same ViT-L backbone and resolution. The training strategy for TopoMLP also mirrors the official.

## B.2 3D-tokenized LLM

**Query Representation.** For the innate priors of the 3D physical world, the query-based BEV framework is introduced. These DETR-style methods, StreamPETR and TopoMLP, extract target-aware query embeddings aka query representations (content) with reference points (localization) to represent objects from multi-view images.

**Reference Point Embeddings.** As previously mentioned, a target is characterized by both its content and location. We integrate the query embeddings by adding reference point embeddings, which are generated from reference points via a single linear layer, to formulate the 3D tokens that represent target information. *A notable aspect of our setup is we initialize the weight of the reference point projector to zero.*

**Memory Queue.** Taking inspiration from StreamPETR, our approach involves the storage of historical queries to preserve continuity in time, as memory queues. Specifically, we concatenate these memory queries with current queries for temporal modeling. To elaborate, our method includes storing queries from three additional frames that exhibit the highest confidence—specifically, the top-K

- "The six images include objects that are uniformly represented as 3D detection query embeddings<query> and map query embeddings<query>. Define the positive y-axis as the forward direction and the positive x-axis as the right direction. The speed of the vehicle is defined as [velocity along the x-axis, velocity along the y-axis]. The acceleration of the vehicle is defined as [acceleration along the x-axis, acceleration along the y-axis]. The ego car will turn left in future. Kindly furnish suitable waypoints for the vehicle's trajectory based on the provided particulars. Waypoints ought to adhere to the [x, y] format, with each waypoint spaced at 0.5-second intervals within a continuous 3.0-second timeframe. For planning tasks, please pay attention to driving safety and avoid vehicle collisions during driving in continous time. "

- "The six images include objects that are uniformly represented as 3D detection query embeddings<query> and map query embeddings<query>. Define the positive y-axis as the forward direction and the positive x-axis as the right direction. The speed of the vehicle is defined as [velocity along the x-axis, velocity along the y-axis]. The acceleration of the vehicle is defined as [acceleration along the x-axis, acceleration along the y-axis]. The ego car will turn right in future. We request your provision of pertinent waypoints for the vehicle's route in accordance with the given information. Waypoints should conform to the format [x, y], with spacing set at 0.5-second intervals over a continuous duration of 3.0 seconds. For planning tasks, please pay attention to driving safety and avoid vehicle collisions during driving in continous time. "

- "The six images include objects that are uniformly represented as 3D detection query embeddings<query> and map query embeddings<query>. Define the positive y-axis as the forward direction and the positive x-axis as the right direction. The speed of the vehicle is defined as [velocity along the x-axis, velocity along the y-axis]. The acceleration of the vehicle is defined as [acceleration along the x-axis, acceleration along the y-axis]. The ego car will go stright in future. Please submit fitting waypoints for the vehicle's course based on the supplied data. Ensure waypoints are structured as [x, y] and spaced at intervals of 0.5 seconds across a continuous 3.0-second period. For planning tasks, please pay attention to driving safety and avoid vehicle collisions during driving in continous time. "

- ......

Table 4: **Question pool of ego planning for VLM-based methods.**

queries, where in our implementation, K is set to 256. The management of these queues adheres to a first-in, first-out (FIFO) principle.

Our 3D-tokenized LLM, Atlas, integrates the 3D tokenizers described earlier with an LLM, specifically the Vicuna-1.5. This LLM has been pre-trained on a diverse open-world data corpus, providing a robust foundation for understanding and processing spatial-temporal. Atlas follows most of the basic settings in Merlin, with a batch size of 1, a learning rate of 2e-5, and the AdamW optimizer with a weight decay of 1e-4. We implement a linear warm-up phase consisting of the first 3% steps in total. Following the warm-up, we transition to a cosine learning rate strategy. The maximum length of prompt tokens is 4096.

## C   3D DETECTION RESULTS

**Precion-Recall Curve.** In the paper text, we present a comparison of the F1 scores between task-specific models and Atlas in 3D detection, focusing on predictions with a confidence score above 0.3, which yielded the highest F1 score. Additionally, we illustrate the performance variations of PETR, StreamPETR, and Atlas through the Precision-Recall curves at different positive thresholds, as shown in Figure 2. It's important to note that Atlas does not generate confidence scores; therefore, we treated all its predictions as positive samples for the purpose of calculating precision and recall. Although Atlas shows slightly weaker performance in making fine-grained predictions (specifically at a threshold of 0.5 meters), it excels in scenarios with larger thresholds. This observation suggests that large language models like Atlas might struggle with highly precise numerical predictions but perform well when broader tolerances are acceptable.

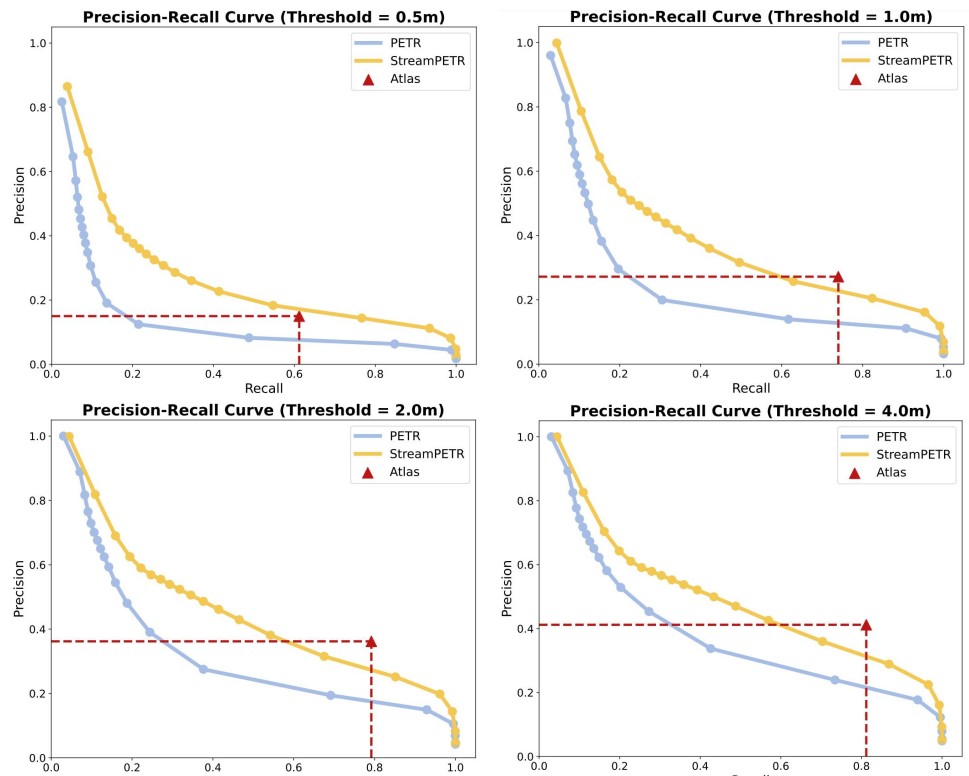

Figure 2: **Compare of 3D detection with various thresholds.** We provide the precision-recall curves of PETR and StreamPETR. As the predictions from Atlas do not include confidence scores, we calculate the precision and recall across all predicted samples.

## D   MORE QUALITATIVE RESULTS

### D.1   QUALITATIVE RESULTS OF 3D DETECTION

We visualize the prediction results of the Atlas model in 3D detection tasks, as shown in Figure 3. The results align well with our performance metrics, demonstrating a notably high recall rate. This high recall is particularly important in practical applications of autonomous driving, where accurately detecting every potential obstacle, like pedestrians, is critical. Furthermore, the model maintains its accuracy even in complex scenarios characterized by high pedestrian density or closely packed targets. Moreover, Atlas proves robust under challenging environmental conditions. For instance, even on rainy days, the model continues to perform strongly. This resilience is essential for the reliability needed in real-world applications, ensuring consistent performance regardless of weather conditions.

### D.2   QUALITATIVE RESULTS OF 3D LANE DETECTION

We showcase the visualization outcomes of Atlas in its application to 3D lane detection, depicted in Figure 4. While the quantitative performance does not surpass task-specific models, Atlas demonstrates noteworthy qualitative performance. As seen, our model performs well in challenging road situations because it accurately recognizes road crossings and dividings.

### D.3   QUALITATIVE RESULTS OF PLANNING

We also demonstrated the adaptability of Atlas's driving plans across various weather conditions in Figure 5. Notably, even during rain, Atlas effectively plans its future travel trajectories with considerable diversity. This capability underscores the model's robustness in challenging environments.

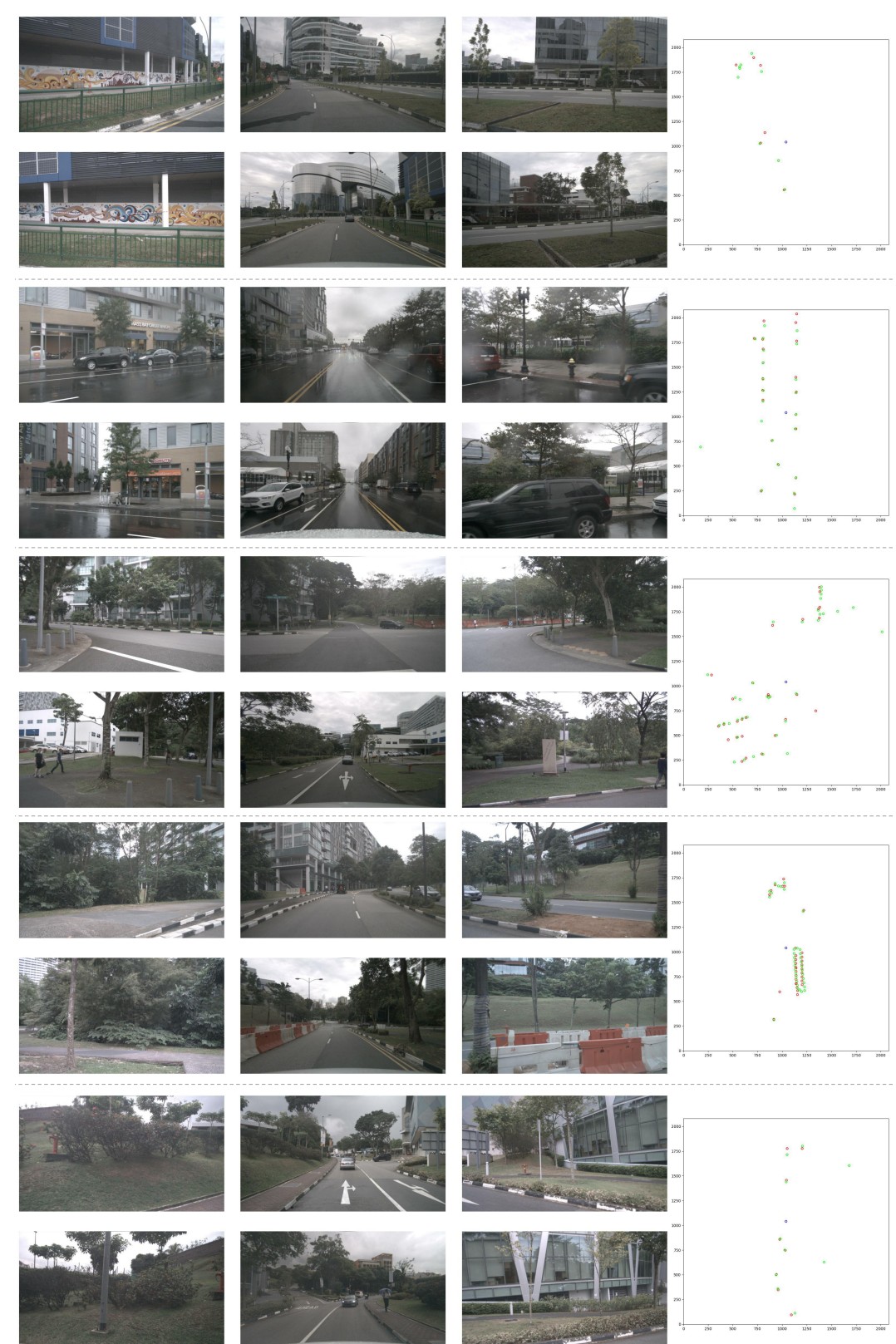

Figure 3: **Qualitative results of Atlas on 3D object detection.** The red circles represent the predicted objects and green circles represent the ground truth.

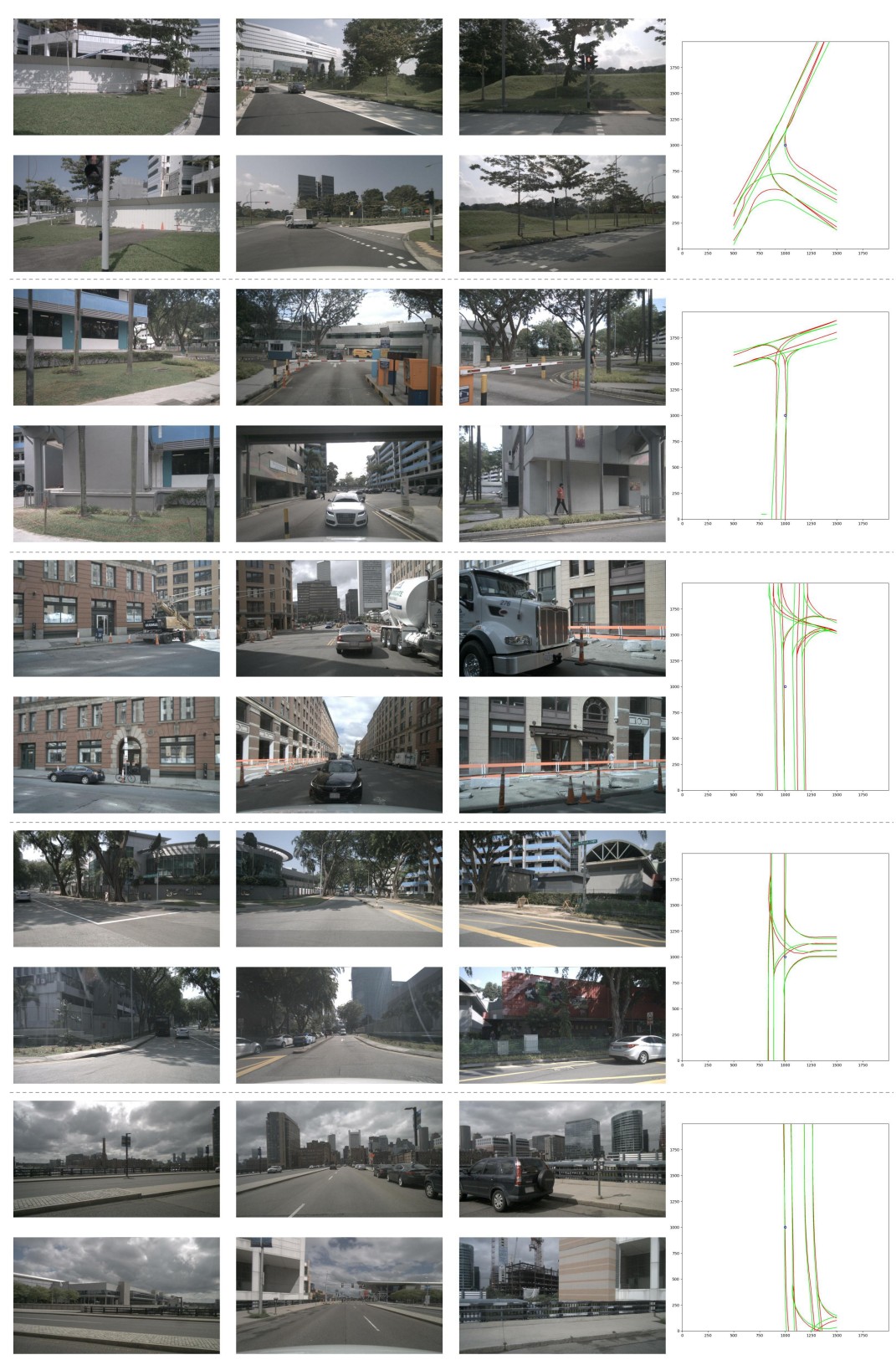

Figure 4: **Qualitative results of Atlas on map detection.** The red represent the predicted lane and green represent the ground truth.

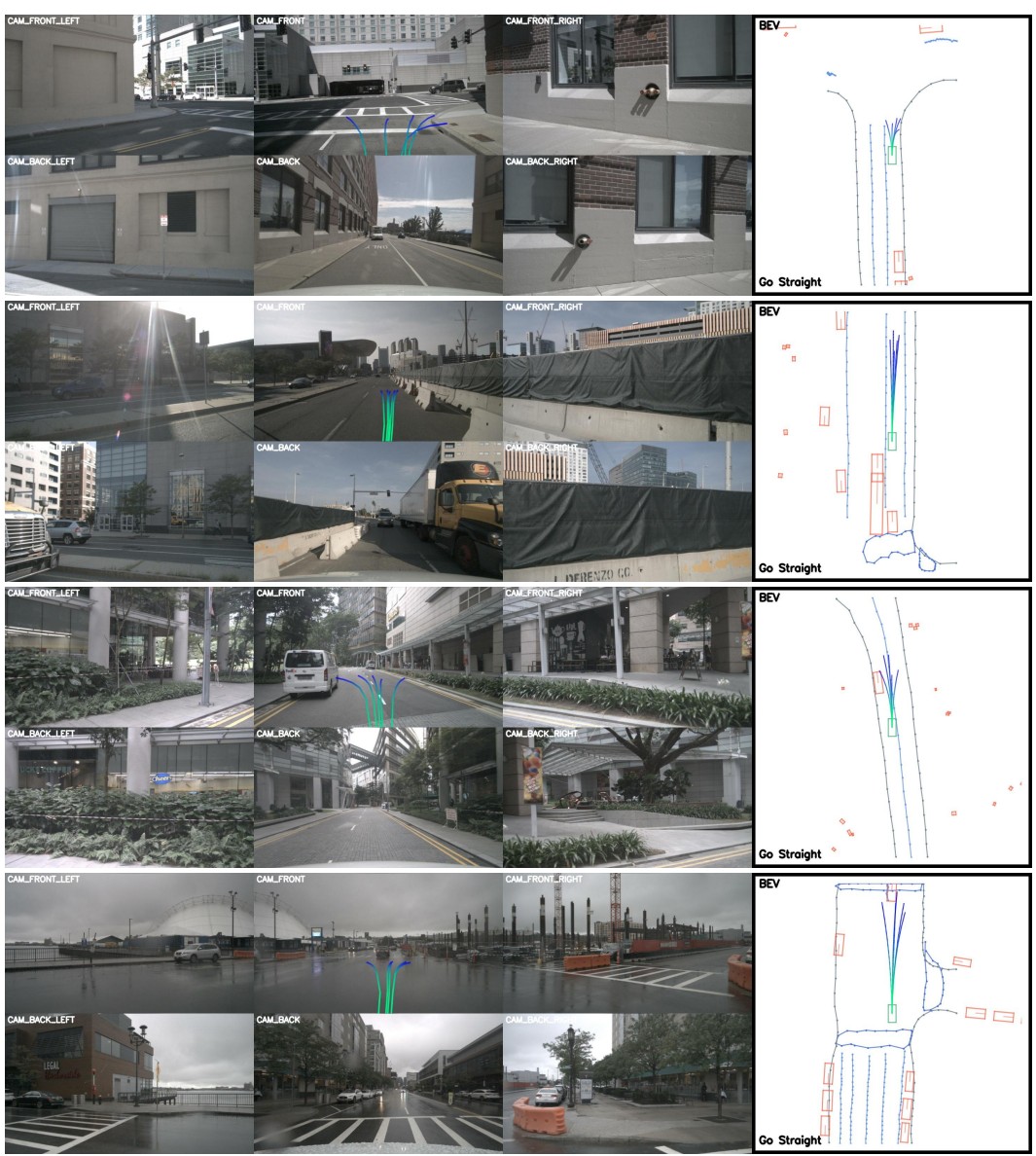

Figure 5: **Qualitative results of Atlas on ego-car planning.** Atlas outputs multiple potential planning trajectories within diverse weather and scenarios.

Furthermore, Atlas impressively maintains compliance with traffic signals, such as stopping at red lights, without having undergone specific training for traffic light recognition. This aspect highlights the model's inherent understanding and application of world knowledge relying on LLM. Additionally, the model's diverse planning strategy enables it to effectively balance the decisions between maintaining its current lane and executing lane changes for overtaking. This flexibility greatly enhances the variety of possible travel routes, adapting dynamically to the flow of traffic and road conditions.

## E    FAILURE CASES

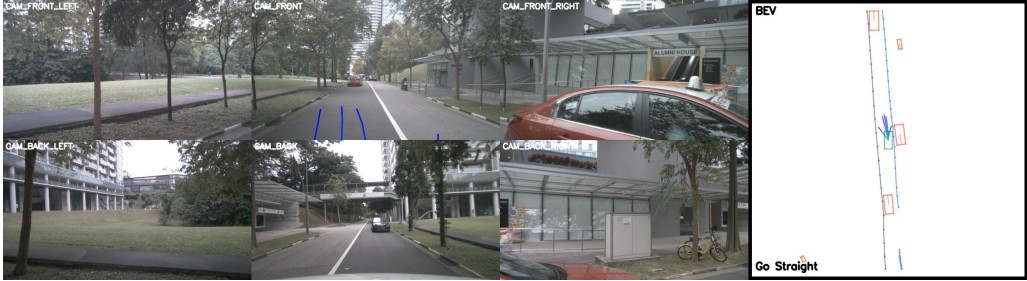

Figure 6: Overly conservative

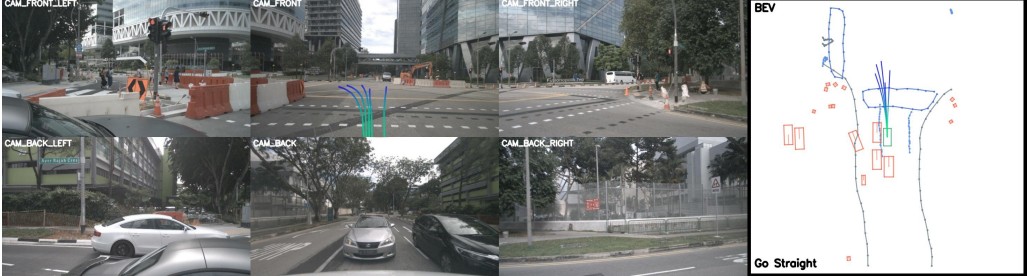

Figure 7: Violation of traffic regulations

Discussing error examples in our model, Atlas, provides valuable insights that can guide future improvements. In this section, we analyze two primary types of failure observed during our experiments:

**Overly Conservative Behavior.** Atlas tends to make overly conservative decisions, favoring caution even when the path ahead is clear, as shown in Figure 6. This behavior results in a lower travel efficiency as the model opts to prioritize safety excessively. Our analysis suggests that this conservatism is likely rooted in the sampling bias of the nuScenes dataset. This dataset predominantly includes safer driving examples and favors lower-speed scenarios, which may have influenced Atlas' decision-making strategy. To address this issue, incorporating a substantial amount of closed-loop data could be beneficial. This would provide Atlas with more dynamic and varied driving scenarios, potentially reducing its overly conservative tendencies.

**Violation of Traffic Regulations.** Despite Atlas having learned to adhere to several traffic rules, it occasionally fails to comply with traffic light signals, as shown in Figure 7. Specifically, Atlas may proceed through intersections during a red light. This error stems from the model's lack of explicit traffic light information in its current framework. To mitigate this issue, integrating enhanced traffic-related data queries could be crucial. By providing Atlas with more explicit and detailed traffic signal information, we can improve its compliance with traffic laws and overall decision-making accuracy.

These findings highlight critical areas for further research and development. Enhancing the dataset and incorporating explicit models of traffic elements such as lights and signs are promising avenues for improving Atlas' performance and reliability.