# OpenReview forum: "Is a 3D-Tokenized LLM the Key to Reliable Autonomous Driving?"
_ICLR.cc/2025/Conference — ICLR 2025 Conference Withdrawn Submission_

### Official Review · Reviewer_ji64 · 2024-10-29

**Soundness:** 3
**Presentation:** 3
**Contribution:** 2
**Rating:** 5
**Confidence:** 4

**Summary:**

This paper combines modular bird’s-eye view (BEV) approaches with vision-language models, integrating 3D priors for enhanced depth perception and supporting multi-view images and temporal modeling via DETR-liked query propagation. Evaluation on the nuScenes dataset showcases notable advancements in 3D object detection, lane detection, and E2E planning, affirming the pivotal role of 3D-tokenized LLMs in ensuring reliable autonomous driving systems.

**Strengths:**

1. The exploration of VLM-based methods in autonomous driving is interesting. This paper highlights the significant differences in perception between task-specific models and 2D tokenized VLM-based approaches.

2. The paper combines DETR-style 3D perceptrons with VLMs, leveraging 3D priors for improved depth perception and supporting various image types and temporal modeling.

3. The evaluation on the nuScenes dataset demonstrates enhancements in 3D detection and planning tasks, showcasing the role of 3D-tokenized VLMs as the key to ensuring reliable autonomous driving.

**Weaknesses:**

1. Overclaiming: The paper argues that existing 2D VLMs struggle in achieving 3D perception, but it has been shown that by pre-training a 2D tokenizer, 2D VLMs can outperform traditional 3D detection methods [1]. The authors are suggested to compare the performance curves of the 2D and 3D tokenizers as the amount of traing data grows. Additionally, other works [2] utilize 3D encoders like depth maps and extrinsics for 3D scene understanding. How does DETR-style tokenizer compare to or improve upon the mothods in [2]?

2. Lack of Innovation:
a) In terms of model designs, some studies [3] have already explored spatial understanding tasks using 3D tokenizers from a BEV perspective. Sec. 3.1's content appears superficial and lacks novelty, making it challenging to discern the paper's contributions. How does Atlas differs from the work in [3], particularly regarding the use of 3D tokenizers.
b) Data preparation, task design, and evaluation metrics mentioned in [1] cover topics discussed in the paper. Could the authors provide a more detailed comparison with [1] to highlight any unique aspects of their work.

3. Based on Fig. 4 and supp Fig. 5, it is not clear whether the model can accurately follow instructions to generate reasonable driving trajectories. Instead, it appears to list all potential trajectories, raising doubts about the model's capability to achieve end-to-end planning tasks. How does the model select the most appropriate trajectory from the multiple options? The authors should provide specific examples or metrics that demonstrate the model's ability to follow instructions.

[1] Zhou, et al. "Embodied Understanding of Driving Scenarios".

[2] Zhu, et al. "LLaVA-3D: A Simple yet Effective Pathway to Empowering LMMs with 3D-awareness".

[3] Choudhary, et al. "Talk2BEV: Language-enhanced Bird's-eye View Maps for Autonomous Driving".

**Questions:**

My concern is the lack of novelty in the methodology of this paper. I hope the authors can compare this work with existing methods to highlight the contributions of this paper.

---

### Official Review · Reviewer_nyTm · 2024-11-01

**Soundness:** 2
**Presentation:** 3
**Contribution:** 2
**Rating:** 5
**Confidence:** 5

**Summary:**

This paper addresses the limitations of current vision-language models (VLMs) used in autonomous driving, which rely primarily on 2D vision tokenizers paired with large language models (LLMs) but lack 3D geometric insights essential for accurate planning. To overcome this gap, the authors propose Atlas, an approach that integrates 3D perception networks as 3D tokenizers with an LLM. Experimental results on the nuScenes dataset indicate that Atlas enhances 3D detection and ego planning, supporting the value of 3D tokenization in advancing reliable autonomous driving.

**Strengths:**

1. The paper is well-written and easy to understand.
2. The work highlights the shortcomings of 2D-tokenized LLMs and demonstrates how a 3D tokenizer can enhance performance.

**Weaknesses:**

1. About the framework design: The necessity of connecting a 3D tokenizer to an LLM is questionable. A baseline experiment could clarify this by showing whether a simple combination of a 3D tokenizer and MLP could achieve similar planning results. The main advantage of incorporating an LLM seems to be interpretability, as LLMs may generate explanations, but whether they can control the trajectory as effectively remains unclear. Additionally, the LLM introduces limitations, such as reduced speed and difficulty in producing accurate trajectory scores. The paper would benefit from discussing the necessity of connecting an LLM to the 3D tokenizer.
2. About the performance: The performance when using the LLM is subpar, especially when ego states are not incorporated. As shown in Table 3, models without ego states underperform significantly compared to BEV-Planner. Conversely, experiments that include ego states are of limited value, as the network may rely on shortcuts derived from ego states rather than genuinely learning to predict trajectories.
3. About the benchmark: The paper lacks experiments on the CARLA simulator, which is critical as the nuScenes dataset alone is inadequate for validation. Although the authors mention that CARLA may lack realism, nuScenes predominantly consists of straightforward driving scenarios, heavily influenced by ego status, making it a limited and potentially biased metric for this work.

**Questions:**

The primary questions are in the weaknesses part noted above. Additional questions include:

1. In Line 157, it is stated, “First, VLMs cannot deliver the necessary predictive confidence for metrics such as mean Average Precision (mAP).” Could the authors clarify the specific challenges associated with generating confidence scores directly for bounding boxes using VLMs? Since VLMs should be capable of producing scores for bounding boxes, are the authors suggesting that these scores lack reliability?
2. To support the claim in Figure 3, are there *quantitative* metrics that substantiate the superior performance of 3D-tokenized VLMs over 2D-tokenized VLMs in driving caption tasks?
3. In Table 3, regarding the collision metric, the authors mention, “We adhere to standard practices by utilizing the implementation provided by ST-P3 (Hu et al., 2022b).” However, the reported results for VAD and UniAD follow the collision metric from BEV-Planner, which yields significantly different values. Could the authors clarify which metric was actually used?

---

### Official Review · Reviewer_9wc5 · 2024-11-04

**Soundness:** 2
**Presentation:** 3
**Contribution:** 3
**Rating:** 5
**Confidence:** 5

**Summary:**

The paper proposes a novel approach for autonomous driving by introducing a 3D-tokenized LLM framework, named Atlas, which integrates a 3D perception mechanism using DETR-style 3D tokenizers (StreamPETR for detection&tracking and TopoMLP for mapping). Traditional VLMs have primarily relied on 2D visual tokenizers, which lack the spatial depth essential for autonomous driving tasks such as 3D object detection, lane perception, and environmental reasoning. Atlas addresses these limitations by leveraging 3D perception models, allowing it to incorporate spatiotemporal data, which significantly enhances environmental understanding and planning tasks. Empirical results on the nuScenes dataset demonstrate that Atlas outperforms previous VLM-based approaches, especially in 3D perception and open-loop planning tasks.

**Strengths:**

- The paper introduces a 3D-tokenized framework that effectively bridges the gap between 2D perception in current VLMs and the 3D requirements of autonomous driving. This integration is both novel and practical for enhancing perception and planning tasks.
- The paper conducts thorough experiments across multiple perception and planning tasks using the nuScenes dataset, with Atlas outperforming previous VLM-based approaches in 3D perception and open-loop planning tasks.
-  The chain-of-thought design for planning is interesting and effective.

**Weaknesses:**

- While this work introduces advancements in spatial perception by incorporating 3D tokenizers, it would benefit from a comparison with recent open-source studies that explore large language models’ spatial understanding in driving scenarios, such as [1].
- The impressive performance in open-loop planning is acknowledged, yet the detailed aspects contributing to this strength remain unclear. Further explanation of the performance-driving factors would enhance the clarity and reliability of these results.
  - Such as, if we use a non-pretrained transformer to replace the LLM, will the results being same?
- Since the 3D tokenizers are trained on labeled driving datasets, solely relying on them could limit the model's adaptability to novel or unseen scenarios, potentially constraining its utility in unpredictable real-world environments.
- To more comprehensively assess environmental understanding, incorporating question-answering tasks, as demonstrated in datasets like DriveLM[2], would provide a well-rounded evaluation of the model’s interpretive abilities.
- In the original 3D tokenizers, object detection relied only on some tiny MLP layers, while the current setup incorporates a more complex LLM, Vicuna. The application of Vicuna following StreamPETR and TopoMLP slightly reduces 3D perception performance.
- Currently, separate tokenizers are used for object detection and lane recognition. For an autonomous driving system that requires multi-tasking, adding tokenizers for each task could pose challenges. Given that each tokenizer includes an image backbone and maybe transformer encoder/decoder, scaling this approach may affect the model's real-time capabilities.

[1] Zhou, Yunsong, et al. Embodied understanding of driving scenarios. In ECCV 2024.
[2] Chonghao, Sima, et al. DriveLM: Driving with Graph Visual Question Answering. In ECCV 2024.

**Questions:**

See Weaknesses.

---

### Official Review · Reviewer_Gg3B · 2024-11-05

**Soundness:** 2
**Presentation:** 3
**Contribution:** 2
**Rating:** 5
**Confidence:** 5

**Summary:**

The paper addresses the limitations of current vision-language models (VLMs) that rely on 2D tokenizers, which struggle with 3D perception essential for reliable autonomous driving. To overcome this, the authors introduce Atlas, a novel framework using 3D tokenizers based on DETR-style 3D perceptrons, allowing enhanced 3D object detection, lane detection, and planning. Evaluated on the nuScenes dataset, Atlas outperforms traditional 2D-tokenized approaches by integrating 3D geometric priors, showcasing its potential for more accurate and reliable end-to-end autonomous driving.

**Strengths:**

- The paper introduces Atlas, a novel 3D-tokenized LLM framework that integrates 3D geometric priors, addressing the limitations of existing 2D-tokenized VLMs in autonomous driving.

- Through comprehensive evaluations on the nuScenes dataset, Atlas demonstrates superior performance in 3D object detection, lane detection, and ego-car planning, validating its effectiveness over traditional methods.

- The paper clearly explains the need for 3D tokenization in autonomous driving, providing structured comparisons and detailed visuals to convey the model’s advantages.

- By advancing reliable end-to-end autonomous driving, this work highlights the essential role of 3D tokenization, with implications for future research in multimodal LLM applications across complex environments.

**Weaknesses:**

- Using 3D perception as tokens will directly benefit from the perception labels, significantly increase the model size, and thus make it **unfair** to compare with those VLMs with images as inputs.

- The paper uses StreamPETR as the 3D perception backbone. However, from Table 1 joint training with LLMs significantly harms the perception performance, which indicates the ineffectiveness of the proposed method.

- Table 2 also shows a performance degradation in lane detection.

- Open-loop planning is not sufficient for evaluating LLM's planning capability. The authors need to evaluate their method in CARLA. Although it is not a real-world simulator, it is still sufficient for just evaluating the closed-loop planning capabilities of the methods

- The paper lacks technical novelty as it just includes explicit 3D perception and lane detection as inputs to LLMs.

**Questions:**

- The authors may also consider evaluating their method on language-based benchmarks such as BDD-X.

---

### Note · Authors · 2024-11-20

I have read and agree with the venue's withdrawal policy on behalf of myself and my co-authors.